# Responses of Root Characteristic Parameters and Plant Dry Matter Accumulation, Distribution and Transportation to Nitrogen Levels for Spring Maize in Northeast China

**Yang Yu** [1,2]**, Chunrong Qian** [2]**, Wanrong Gu** [1,2] **and Caifeng Li** [1,2,*]

[1]  College of Agriculture, Northeast Agricultural University, Harbin 150030, China; yangyu@haas.cn (Y.Y.); wanronggu@neau.edu.cn (W.G.)
[2]  Institute of Crop Cultivation and Tillage, Heilongjiang Academy of Agricultural Sciences, Harbin 150086, China; qianchunrong@haas.cn
[*]  Correspondence: licaifeng@neau.edu.cn; Tel.: +86-451-551-90472

**Abstract:** Improving nitrogen use efficiency is a significant scientific problem to be solved. Two maize hybrids JD27 (Jidan 27) and SD19 (Sidan 19) were selected to study the effects of nitrogen levels on root characteristic parameters and plant dry matter accumulation, distribution and transportation. We set five different nitrogen levels, which were nitrogen deficiency (000N), low nitrogen (075N), medium nitrogen (150N), high nitrogen (225N) and excessive nitrogen (300N). The results showed that the root length and root surface area of JD27 were significantly higher than those of SD19 under 075N. With the increase of nitrogen levels, the root difference among varieties gradually decreased. The root length, projection area, total surface area and total volume reached the maximum values at silking stage. The average root diameter kept stable or decreased slowly with the growth stage. The dry matter accumulation of JD27 was higher than that of SD19 at all growth stages. Increasing the amount of nitrogen fertilizer can promote the transport of dry matter to grain and improve dry matter transport efficiency after anthesis. Under the treatment of medium and high nitrogen fertilizer, maize was easy to obtain a higher yield, but excessive nitrogen fertilizer inhibited the increase of yield. This study provides theoretical and practical guidance for maize production techniques.

**Keywords:** maize; nitrogen level; root characteristic parameters; dry matter accumulation; distribution and transport; yield

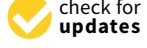



## 1. Introduction

Nitrogen is an essential factor affecting plant growth and development, yield and quality, and plays an important role in plant life's physiological regulation [1]. Nitrogen fertilizer is the most widely used in maize production [2,3]. Since the 1970s, the world's grain production has doubled and the amount of nitrogen fertilizer has increased seven times, but the average nitrogen use efficiency (NUE) of cereal crops in the world is only 33%. According to relevant literature, the amount of nitrogen fertilizer applied to fields in Laizhou City, Shandong Province, China, was as high as 993 kg ha$^{-1}$ [4,5]. Maize nitrogen use efficiency is a quantitative trait controlled by multiple genes, so it must be expressed through a series of morphological, physiological and biochemical characteristics [6,7]. Root growth is closely related to physiological metabolism and dry matter accumulation of shoots [8,9]. When the structure and function of root and crown are in balance, their growth proportion is coordinated, and their yield and resource utilization efficiency are high [10]. The difference of nitrate utilization ability of maize varieties not only depends on the amount of nitrogen absorbed by aboveground parts, but also depends on the root growth [11,12]. The root morphology, distribution and physiological and biochemical characteristics have significant effects on nitrogen absorption. The difference in nitrogen use efficiency was mainly related to nitrogen absorption and dry matter production. The

genotypic difference of nitrogen efficiency mainly came from utilization efficiency under low nitrogen, and mainly from absorption efficiency under high nitrogen [13]. Modern varieties showed higher relative growth rate of stem and leaf under high nitrogen and low nitrogen levels [14].

　　　Maize growth is the process increasing dry matter accumulation, which can be divided into three stages: vegetative growth stage, vegetative growth and reproductive growth stage, and reproductive growth stage. With the development of growth process, the dry matter accumulation of maize presents an "S" curve trend [15,16]. The logistic equation is often used to fit this trend [17–19]. Dry matter accumulation, distribution and translocation affect maize yield and grain quality [20,21]. Nitrogen levels can increase dry matter accumulation and grain transport efficiency, which is the primary method to obtain high maize yield [22,23]. Previous studies showed that dry matter production of maize entered a rapid growth period after the jointing stage, and the dry matter accumulation rate was significantly accelerated [24]. In a specific range, nitrogen can dramatically increase the dry matter accumulation of maize, but excessive nitrogen levels can reduce the dry matter accumulation [25,26]. Fractional nitrogen levels could significantly increase the dry matter accumulation of maize population and promote dry matter transport before anthesis and the accumulation of dry matter after anthesis [27,28]. Previous experiments on nitrogen fertilizer and density setting of maize were mainly discussed from the perspectives of maize leaf canopy and yield. Based on the background of "increasing density and reducing nitrogen" of maize, this study focused on Heilongjiang maize in the main production area of China, mainly from the changes of characteristic parameters of the underground root system, as well as the accumulation and transportation of dry matter. This study will reveal the response of root system to nitrogen and the coordination relationship between root system and shoot under different densities, which can not only ensure the high and stable yield of maize, but also significantly improve the nitrogen use efficiency of maize. In this study, two maize hybrids with significant differences in nitrogen use efficiency were selected as materials to reveal the root characteristic parameters and the response characteristics of plant dry matter accumulation, distribution and transportation to different nitrogen levels, which has important theoretical and practical significance for accelerating the cultivation of nitrogen-efficient maize varieties and the innovation of cultivation technology. Therefore, based on the selection of excellent maize varieties, we believe that the synergy of maize yield increase and nitrogen fertilizer efficiency can be achieved through the innovation of cultivation technology and mode integration.

## 2. Materials and Methods

### 2.1. Experimental Sites and Varieties

　　　Two maize varieties with a significant difference in nitrogen use efficiency were tested, namely, Jidan 27 (JD27) with low-nitrogen use efficiency and Sidan 19 (SD19) with high-nitrogen use efficiency [29]. The growth period of the two varieties was 118–120 days. From 2015 to 2016, the experiment was carried out in the modern agriculture demonstration area of Heilongjiang Academy of Agricultural Sciences in Harbin City (45°50′54″ N, 126°50′12″ E), Heilongjiang Province, China. The soil basic fertility values were as follows—total nitrogen (0.47 g kg$^{-1}$), total phosphorus (0.23 g kg$^{-1}$), total potassium (2.71 g kg$^{-1}$), organic matter (1.87 g kg$^{-1}$), alkali hydrolysable nitrogen (216.55 mg kg$^{-1}$), available phosphorus (57.22 mg kg$^{-1}$), available potassium (125.17 mg kg$^{-1}$) and pH 6.5. Average temperature, precipitation and sunshine hours of the 2015 and 2016 experimental seasons are shown in Table 1.

**Table 1.** Daily mean values of the weather variables at the experimental site during the six months of the maize growing season in 2015 and 2016.

| Month | Average Temperature (°C) | | Precipitation (mm) | | Sunshine Hours (h) | |
|---|---|---|---|---|---|---|
| | 2015 | 2016 | 2015 | 2016 | 2015 | 2016 |
| April | 15.4 | 8.0 | 75.4 | 15.2 | 223.2 | 219.10 |
| May | 21.8 | 16.0 | 33.8 | 106.8 | 274.6 | 183.00 |
| June | 25.2 | 20.1 | 45.7 | 206.1 | 231.4 | 238.10 |
| July | 30.4 | 24.3 | 20.4 | 44.2 | 286.7 | 246.20 |
| August | 26.6 | 23.2 | 65.1 | 31.7 | 200.1 | 283.70 |
| September | 22.3 | 17.1 | 43.2 | 70.3 | 207.3 | 152.40 |
| Total | 23.6 | 18.1 | 283.6 | 474.3 | 1423.3 | 1322.50 |

*2.2. Experimental Design*

Five pure nitrogen levels were set up in the experiment, which were named 0 kg ha$^{-1}$ (000N) of nitrogen deficiency treatment, 75 kg ha$^{-1}$ (075N) of low nitrogen level, 150 kg ha$^{-1}$ (150N) of medium nitrogen level, 225 kg ha$^{-1}$ (225N) of high nitrogen level and 300 kg ha$^{-1}$ (300N) of excessive nitrogen level. The seeds were mechanically graded and selected before sowing. The seeds were coated 2 days before sowing, and the seed coating agent containing carbofuran or tebuconazole was selected for coating. Two treatments conducted weed control in maize field, namely, seedling closed soil treatment and spray treatment. Biological measures and chemical pesticide measures were used to control maize borer and big leaf spot. Other agronomic measures refer to local high-yield cultivation measures. A split plot experiment design was adopted, with three repetitions. The main plot was nitrogen fertilizer treatment, and the variety was a split plot. Each district has 6 rows, 8 m in length and 0.65 m in row spacing. The field planting density was 60,000 plants ha$^{-1}$. Nitrogen fertilizer was applied twice, 1/3 of which was applied at sowing, the remaining 2/3 was applied at V6 (6 leaf development stage), and phosphorus fertilizer and potassium fertilizer were applied at one time at sowing. The application of P$_2$O$_5$ and K$_2$O were 75 kg ha$^{-1}$ and 75 kg ha$^{-1}$, respectively.

*2.3. Measurement and Methods*

2.3.1. Sampling and Measurements

Methods of root extraction were as follows. Take three plants from each plot in V13 (13 leaf development stage), R1 (silking stage), R3 (milk stage) and R5 (dent stage). Take the plant as the center, draw a square with side length of 26 cm as the root area, and take the root depth of 30 cm. Dig out the earth with roots and put it into the mesh bag, soak it in the pool for 30~60 min, wash away most of the soil, and then wash it further with the washing machine until there is no soil residue, remove impurities, cut off the branch roots from the main roots, put it into the self-sealing bag together with the scattered fibrous roots, and put it in the 4 °C fresh-keeping cabinet for indoor measurement of root indexes.

2.3.2. Determination of Root Characteristic Parameters

The root indexes were measured by the wanshen LA-S root analysis system. This system is used for automatic analysis of root images with more than 0.2 mm diameter after cleaning. The pretreated roots were evenly placed in the root disk with backlight, the root disk was made of acrylic material, the specification was length × width × height = 60 cm × 60 cm × 7 cm, the root disk was placed horizontally and added with appropriate amount of water, and the root disk was photographed vertically with an SLR (single-lens reflex) camera to obtain root images. The root images were imported into the analysis system to obtain root length, average root diameter, root projection area, root pixel area, total root surface area and total root volume. The specific index calculation method is as follows.

Root length: the Euclidean distance between every two adjacent pixels is calculated along the root axis and summed.

Average root diameter: along the central axis, calculate the nearest distance (radius) from each pixel to the background point and calculate the average value, average diameter = 2 × average radius;

Root projection area: average diameter × length of root;

Root pixel area: the sum of the area of all pixels (small squares) of the root;

Total root surface area: average root diameter × π × root length;

Total root volume: average diameter × average diameter × length × 0.25 × π

### 2.3.3. Determination of Characteristics of Dry Matter Accumulation, Distribution and Transportation in Plants

In V7 (7 leaf development stage), V13 (13 leaf development stage), R1 (silking stage), R3 (milk stage), R5 (dentate stage), five plants were taken from each plot, and divided and bagged according to stem leaf sheath, leaf, tassel, petiole axis bract, grain and other organs. After drying to constant weight, the dry matter accumulation, distribution and transportation were calculated.

### 2.3.4. Logistic Fitting of Plant Dry Matter

The growth rate of plant organs will show the basic law of "slow fast slow", which is called the grand period of growth. Generally, the logistic equation of "S" curve is used to fit the changes of dry weight, height, surface area, cell number and other traits of crops [30,31]. Through slogistic3 function of logistic equation, we can get the following data, such as final theoretical biomass yield (a), maximum growth rate (Vmax), average growth rate (Va), Time of instantaneous maximum slope (Tm), Time of growth rate acceleration (T1), Time of growth rate deceleration ($T_2$) and rapid growth rate period ($T_{2-1}$) [32,33].

$$y = \frac{a}{1 + b e^{-kx}}.$$

In the equation, y is the dry matter accumulation of maize; a, B and K are the parameters of logistic equation in the process of dry matter accumulation; a is the relative potential maximum dry matter accumulation potential; B is the retardation coefficient; K is the growth rate of relative dry matter; X is the relative growth days; E is the natural constant. In this study, slogistic3 function in growth/sigmoidal category in "analysis fitting nonlinear curve fitting" dialog box under origin 2018 software was used for fitting calculation [34].

### 2.3.5. Relative Indexes of Plant Dry Matter Accumulation and Transportation

Five plants with uniform growth were selected in each plot at each growth stage, which were killed at 105 °C for 30 min and dried to constant weight at 80 °C. The dry matter accumulation was determined, and the dry matter accumulation and contribution rate of single plant after flowering were calculated according to the following equation [35–39].

Dry matter accumulation (DMA, kg ha$^{-1}$) = dry matter accumulation $T_2$ − matter accumulation $T_1$;

Dry matter accumulation rate (DMAR, kg ha$^{-1}$ d$^{-1}$) = (dry matter accumulation $T_2$ − matter accumulation $T_1$)/ ($T_2$ − $T_1$);

Harvest index (HI, %) = dry weight of R6 grain/total aboveground dry matter accumulation of R6 plant

Redistribution amount of pre silking stored dry matter vegetative organs to grain (RAP, t ha$^{-1}$) = dry weight of vegetative organs before anthesis (R1) − dry weight of vegetative organs after anthesis (R5);

Percentage of redistributed pre silking stored dry matter from vegetative organs to grain (PRAP, %) = dry matter transport amount before anthesis/dry matter weight of vegetative organs before anthesis (R1);

$$\text{Contribution of RAP to grain yield (CRAP, \%) = dry matter}$$
$$\text{transport amount before anthesis/dry weight of grain after anthesis (R5)} \times 100;$$

$$\text{Amount of post anthesis transfer of assimilated dry matter into grain (APA, t ha}^{-1}) = \text{dry weight of grain}$$
$$\text{after anthesis (R5)} - \text{assimilate transfer of vegetative organs before anthesis (R1);}$$

$$\text{Contribution of APA to grain matter (CAPA, \%) = dry matter assimilation/dry weight of grain}$$
$$\text{after anthesis (R5)} \times 100$$

### 2.3.6. Determination of Yield and Its Components

At the mature stage, the middle two rows of each plot were harvested. After air drying, five representative ears were selected to investigate the ear length, ear diameter, bald tip length, row number and grain number. After the investigation of ear characters, the ear weight, grain weight and water content of grain were determined by the yield measurement system of Dongsheng grain plot in Tieling, China. 200 g grain was taken evenly and dried for 100 grain weight determination [40–42].

### 2.4. Data Analysis

According to the analysis of variance, data were statistically analysed following standard methods using Microsoft Excel 2010 and SPSS 12.0. Differences between treatments were determined by a posteriori Tukey's test at a significance level of 0.05 and 0.01.

## 3. Results

### 3.1. Apparent Characteristics of Root System

Figure 1 shows the apparent overall trend of the maize root system under different nitrogen levels in V13-R3 period. With the growth process of maize, the root volume and the number of roots gradually increase. Also, the length of main roots and the amount of fibrous roots increases, and the root sensory performance becomes denser. With the increase of nitrogen levels, the root diameter gradually increased, and the sensory performance showed that the root system gradually became solid and fibrous roots became dense gradually. Compared with V13 stage, the sensory performance of root density in R1 stage increased slightly but the difference was not obvious, which indicated that the main root differentiation might be completed in V13-R1 stage. The root density in R3 stage was significantly higher than that of in V13 and R1 stages, and the number of fibrous roots increased significantly, which indicated that the root growth in the later stage of growth and development might be dominated by fibrous root differentiation. In V13 stage, there was no significant difference in root volume between JD27 and SD19 under nitrogen levels. In R1 stage, the root volume of JD27 was larger than that of SD19 under 000N, 075N and 150N, but there was no significant difference between JD27 and SD19 under 225N and 300N, indicating that the root growth of JD27 was better than that of SD19 under medium and low nitrogen. In R3 stage, the overall sensory concentration of JD27 under nitrogen levels was better than that of SD19, indicating that the fibrous root amount of JD27 may be more than that of SD19. Under 225N and 300N, the two varieties' sensory performance was basically the same, which indicated that excessive nitrogen levels did not necessarily increase the root weight of maize at the later growth stage.

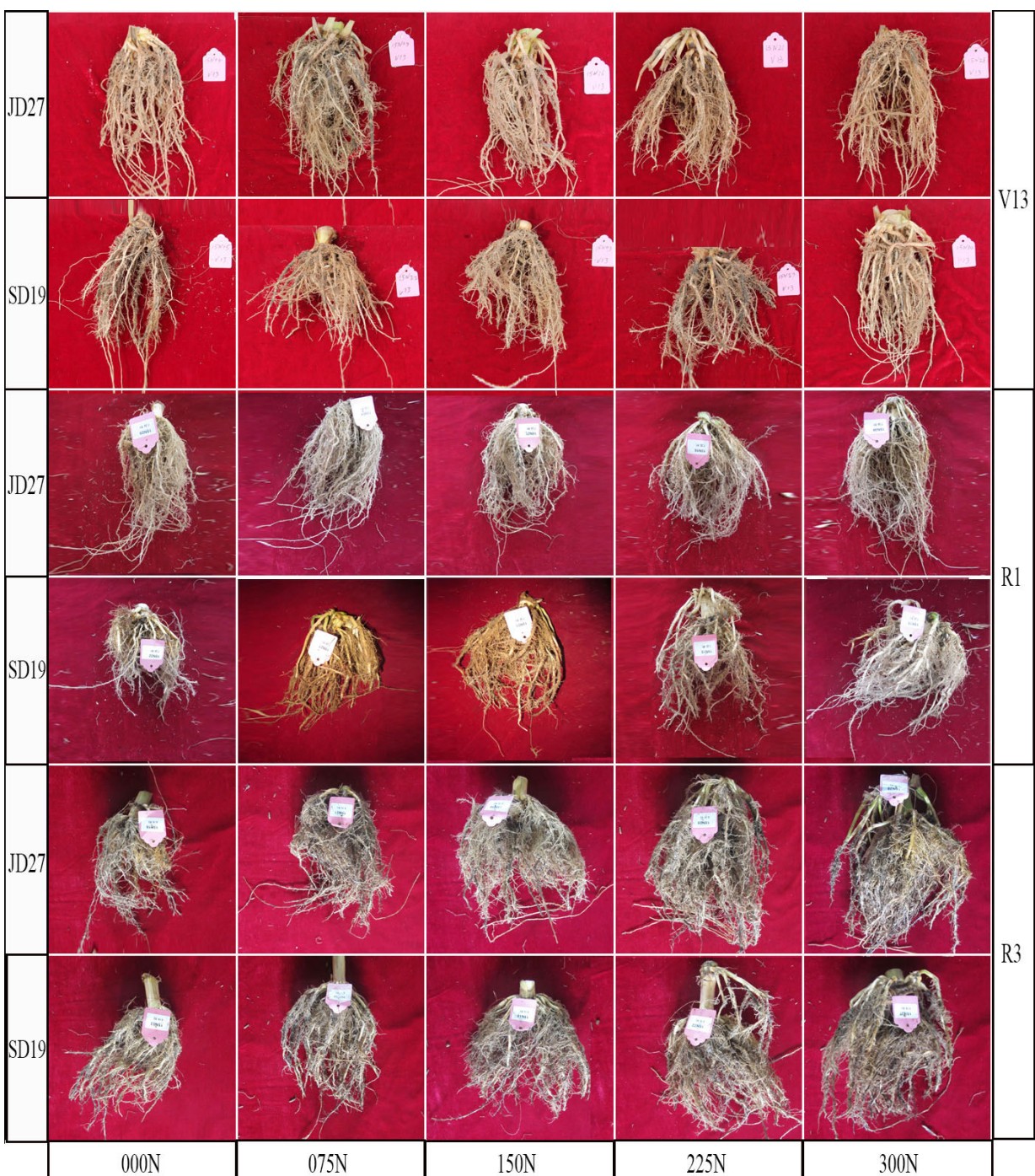

**Figure 1.** Apparent root characteristics of maize varieties under nitrogen levels within V13–3 stage.

### 3.2. Root Length

Factor analysis of variance showed that the root lengths of JD27 and SD19 were significantly different ($p < 0.01$) except for V13 and R5. Under the interaction of varieties and nitrogen levels, the root length was significantly different ($p < 0.01$) except at V13 stage. The root length of JD27 and SD19 increased first and then decreased with the increase of nitrogen levels. In V13 and R1 stage, the root length of JD27 under 075N was significantly higher than that of 000N. For JD27, the root length of 150N was 2.49% less than that of 000N, and that of 300N was 2.69% less than that of 000N. For SD19, the root length of 150N was 22% more than that of 000N, and that of 300N was 22.00% more than that of 000N. With the increase of nitrogen levels, the root length change of SD19 was significantly higher

than that of JD27, indicating that JD27 was more advantageous than SD19 in root length under low nitrogen condition, and high nitrogen level was more conducive to the increase of root length of SD19 (Table 2).

**Table 2.** Effects of nitrogen levels on root length of maize varieties.

| Varieties | Nitrogen Levels | V13 (m plant$^{-1}$) | R1 (m plant$^{-1}$) | R3 (m plant$^{-1}$) | R5 (m plant$^{-1}$) |
|---|---|---|---|---|---|
| JD27 | 000N | 16.87 c | 29.52 cd | 46.32 ab | 45.72 a |
| JD27 | 075N | 29.53 a | 35.72 ab | 44.51 ab | 42.40 a |
| JD27 | 150N | 30.19 a | 32.82 abc | 39.22 b | 32.75 b |
| JD27 | 225N | 31.80 a | 38.39 a | 44.17 ab | 32.25 b |
| JD27 | 300N | 27.35 a | 30.38 bc | 48.40 a | 28.58 b |
| SD19 | 000N | 19.83 bc | 24.20 de | 26.46 c | 29.85 b |
| SD19 | 075N | 20.50 bc | 20.99 e | 26.79 c | 26.07 b |
| SD19 | 150N | 27.14 a | 27.19 cd | 39.18 b | 28.91 b |
| SD19 | 225N | 30.98 a | 37.54 a | 48.36 a | 42.16 a |
| SD19 | 300N | 25.15 ab | 37.29 a | 44.75 ab | 46.21 a |
| F value | Varieties (V) | 3.56 | 8.81 ** | 18.73 ** | 0.97 |
| F value | Nitrogen (N) | 11.57 ** | 9.26 ** | 7.62 ** | 2.35 |
| F value | V:N | 2.28 | 7.11 ** | 7.91 ** | 15.55 ** |
| JD27 | | 27.15 a | 33.37 a | 44.52 a | 36.34 a |
| SD19 | | 24.72 a | 29.44 b | 37.11 b | 34.64 a |
| | 000N | 18.35 c | 26.86 c | 36.39 b | 37.78 a |
| | 075N | 25.02 b | 28.36 c | 35.65 b | 34.24 ab |
| | 150N | 28.66 ab | 30.01 bc | 39.20 b | 30.83 b |
| | 225N | 31.39 a | 37.97 a | 46.26 a | 37.21 a |
| | 300N | 26.25 b | 33.83 ab | 46.58 a | 37.39 a |

Different lowercase letters are significant difference at $p < 5\%$ on treatment, ** $p < 0.01$.

### 3.3. Average Root Diameter

Factor analysis of variance showed that there was no significant difference in root diameter between JD27 and SD19 at V13, but the difference was extremely significant ($p < 0.01$) at R1, R3 and R5. Under the interaction of varieties and nitrogen levels, there was no significant difference in average root diameter at R1 and R3, but the difference was extremely significant at V13 and R5 ($p < 0.01$). There was no significant difference in root diameter at V13 and R1, but significant difference at R3 and R5 ($p < 0.05$). For JD27, the average root diameter increased by 1.06% at 150N and 16.11% at 300N. For SD19, the average root diameter increased by 1.06% at 150N and 16.11% at 300N. The average root diameter increased by 0.45% at 300N compared with 000N. The results showed that SD19 was superior to JD27 in average root diameter under high nitrogen condition, and the increase or decrease of nitrogen levels had little effect on the average root diameter of JD27 (Table 3).

### 3.4. Root Projection Area

Factor analysis of variance showed that the root projection areas of JD27 and SD19 were significantly different at R1 stage ($p < 0.01$), but there were no significant differences in V13, R3 and R5. Under the interaction of varieties and nitrogen levels, the root projection area had no significant difference at V13, reached significant difference at R3 ($p < 0.05$), and reached an extremely significant difference at R1 and R5 ($p < 0.01$). There was no significant difference in root projection area at V13 stage under different nitrogen levels, but the difference was significant ($p < 0.05$) at R5 stage, and extremely significant ($p < 0.01$) at R1 and R3 stages. With the increase of nitrogen levels, the root projection area increased significantly, but the difference was different for different varieties. For SD19, the root projection area increased by 29.89% at 150N and 300N, and the area increased by 60.16%. Under 000N and 075N, the change of root projection area of JD27 was significantly higher than that of SD19. Under 225N and 300N, the change of root projection area of SD19 was

significantly higher than that of JD27, which indicated that JD27 was more conducive to the increase of root projection area than SD19 under low nitrogen cultivation condition, and high nitrogen level was more conducive to the increase of root projection area of SD19 (Table 4).

**Table 3.** Effects of nitrogen levels on root mean diameter of maize varieties.

| Varieties | Nitrogen Levels | V13 (mm plant$^{-1}$) | R1 (mm plant$^{-1}$) | R3 (mm plant$^{-1}$) | R5 (mm plant$^{-1}$) |
|---|---|---|---|---|---|
| JD27 | 000N | 1.39 d | 1.58 abc | 1.38 de | 1.30 cd |
| JD27 | 075N | 1.47 d | 1.67 ab | 1.30 e | 1.74 a |
| JD27 | 150N | 1.55 bcd | 1.32 c | 1.51 cde | 1.33 cd |
| JD27 | 225N | 1.68 abcd | 1.48 bc | 1.77 abcd | 1.24 d |
| JD27 | 300N | 1.79 abc | 1.59 abc | 1.70 abcde | 1.48 bc |
| SD19 | 000N | 1.69 abcd | 1.67 ab | 1.92 abc | 1.35 cd |
| SD19 | 075N | 1.93 a | 1.80 a | 1.54 bcde | 1.46 cd |
| SD19 | 150N | 1.44 d | 1.85 a | 1.62 bcde | 1.75 a |
| SD19 | 225N | 1.86 ab | 1.70 ab | 2.08 a | 1.72 ab |
| SD19 | 300N | 1.54 cd | 1.57 abc | 2.00 ab | 1.55 abc |
| F value | Varieties (V) | 3.63 | 9.55 ** | 11.16 ** | 8.29 ** |
| F value | Nitrogen (N) | 2.72 | 0.89 | 4.17 * | 3.36 * |
| F value | V:N | 4.45 ** | 2.41 | 0.59 | 7.44 ** |
| JD27 | | 1.58 a | 1.53 b | 1.53 b | 1.42 b |
| SD19 | | 1.69 a | 1.72 a | 1.83 a | 1.57 a |
| | 000N | 1.54 bc | 1.62 a | 1.65 abc | 1.32 b |
| | 075N | 1.70 ab | 1.73 a | 1.42 c | 1.60 a |
| | 150N | 1.50 c | 1.59 a | 1.57 bc | 1.54 a |
| | 225N | 1.77 a | 1.59 a | 1.93 a | 1.48 ab |
| | 300N | 1.67 abc | 1.58 a | 1.85 ab | 1.52 a |

Different lowercase letters are significant difference at $p < 5\%$ on treatment, ** $p < 0.01$, * $p < 0.05$.

**Table 4.** Effects of different nitrogen levels on root projection area of maize varieties.

| Varieties | Nitrogen Levels | V13 (cm$^2$ plant$^{-1}$) | R1 (cm$^2$ plant$^{-1}$) | R3 (cm$^2$ plant$^{-1}$) | R5 (cm$^2$ plant$^{-1}$) |
|---|---|---|---|---|---|
| JD27 | 000N | 414.90 abc | 435.44 cd | 589.17 cde | 540.77 bcd |
| JD27 | 075N | 416.39 abc | 543.15 bc | 695.11 bcd | 568.89 abc |
| JD27 | 150N | 409.05 bc | 411.34 d | 615.46 cde | 568.86 abc |
| JD27 | 225N | 368.38 bc | 420.75 d | 790.07 b | 507.58 cd |
| JD27 | 300N | 409.85 bc | 450.69 cd | 716.28 bc | 565.11 bc |
| SD19 | 000N | 316.98 c | 388.13 d | 503.72 e | 387.25 d |
| SD19 | 075N | 413.14 bc | 460.82 cd | 550.26 de | 433.14 cd |
| SD19 | 150N | 459.43 abc | 503.48 bcd | 624.27 bcde | 485.95 cd |
| SD19 | 225N | 580.15 a | 792.70 a | 971.57 a | 725.78 a |
| SD19 | 300N | 518.91 ab | 603.95 b | 747.08 bc | 686.36 ab |
| F value | Varieties (V) | 2.66 | 15.26 ** | 0.00 | 0.05 |
| F value | Nitrogen (N) | 1.37 | 6.97 ** | 12.67 ** | 4.35 * |
| F value | V:N | 2.47 | 10.57 ** | 2.91 * | 5.90 ** |
| JD27 | | 403.71 a | 452.27 b | 681.22 a | 550.24 a |
| SD19 | | 457.72 a | 549.82 a | 679.38 a | 543.70 a |
| | 000N | 365.94 a | 411.79 c | 546.44 c | 464.01 b |
| | 075N | 414.77 a | 501.99 b | 622.69 bc | 501.01 b |
| | 150N | 434.24 a | 457.41 bc | 619.86 bc | 527.41 ab |
| | 225N | 474.26 a | 606.73 a | 880.82 a | 616.68 a |
| | 300N | 464.38 a | 527.32 ab | 731.68 b | 625.74 a |

Different lowercase letters are significant difference at $p < 5\%$ on treatment, ** $p < 0.01$, * $p < 0.05$.

### 3.5. Root Pixel Area

Factor analysis of variance showed that there was no significant difference in root pixel area of JD27 and SD19 in each period. Under the interaction of varieties and different nitrogen levels, the root pixel area was significantly different ($p < 0.01$) except for V13 and R3. There was no significant difference in root pixel area at R5, but significant difference at V13 ($p < 0.05$), and extremely significant difference at R1 and R3 ($p < 0.01$). With the increase of nitrogen levels, the root pixel area of JD27 decreased first and then increased, while that of SD19 increased obviously. However, the difference of variation amount was different for different varieties, and the difference was significant for JD for SD19, the pixel area of roots at different growth stages of 150N decreased by 2.21% compared with that of 000N, and that of roots at 300N increased by 2.99% compared with that of 000N. For SD19, the pixel area of roots at different growth stages of 150N increased by 22.54% compared with that of 000N. Under 000N and 075N, the change of root pixel area of JD27 was significantly higher than that of SD19. However, under 225N and 300N, the change of root pixel area of SD19 was significantly higher than that of JD27, which indicated that JD27 was more conducive to the increase of root pixel area than SD19 under low nitrogen level, and high nitrogen level was more conducive to the increase of root pixel area of SD19 (Table 5).

**Table 5.** Effects of nitrogen levels on root pixel area of maize varieties.

| Varieties | Nitrogen Levels | V13 (cm$^2$ plant$^{-1}$) | R1 (cm$^2$ plant$^{-1}$) | R3 (cm$^2$ plant$^{-1}$) | R5 (cm$^2$ plant$^{-1}$) |
|---|---|---|---|---|---|
| JD27 | 000N | 296.61 bc | 387.75 bc | 400.52 cd | 396.58 abc |
| JD27 | 075N | 309.25 bc | 391.30 b | 388.51 cd | 446.02 ab |
| JD27 | 150N | 319.96 abc | 326.19 bcd | 402.48 cd | 399.98 abc |
| JD27 | 225N | 348.98 ab | 399.83 b | 516.84 ab | 291.19 d |
| JD27 | 300N | 320.99 abc | 331.98 bcd | 487.13 bc | 385.59 bc |
| SD19 | 000N | 231.46 c | 268.01 d | 332.96 d | 296.26 d |
| SD19 | 075N | 270.39 bc | 286.97 cd | 383.70 d | 284.69 d |
| SD19 | 150N | 312.32 bc | 324.07 bcd | 423.99 bcd | 322.73 cd |
| SD19 | 225N | 416.15 a | 556.02 a | 608.33 a | 479.48 a |
| SD19 | 300N | 341.75 ab | 406.32 b | 495.58 bc | 473.94 ab |
| F value | Varieties (V) | 0.05 | 0.00 | 0.22 | 0.48 |
| F value | Nitrogen (N) | 3.79 * | 8.58 ** | 12.11 ** | 2.56 |
| F value | V:N | 1.25 | 7.23 ** | 1.48 | 12.98 ** |
| JD27 | | 319.16 a | 367.41 a | 439.10 a | 383.87 a |
| SD19 | | 314.41 a | 368.28 a | 448.91 a | 371.42 a |
| | 000N | 264.04 b | 327.88 b | 366.74 c | 346.42 b |
| | 075N | 289.82 b | 339.13 b | 386.10 c | 365.36 b |
| | 150N | 316.14 ab | 325.13 b | 413.23 c | 361.36 b |
| | 225N | 382.57 a | 477.92 a | 562.58 a | 385.33 ab |
| | 300N | 331.37 ab | 369.15 b | 491.35 b | 429.76 a |

Different lowercase letters are significant difference at $p < 5\%$ on treatment, ** $p < 0.01$, * $p < 0.05$.

### 3.6. Total Root Surface Area

Factor analysis of variance showed that the total root surface area of JD27 and SD19 only reached the extremely significant difference ($p < 0.01$) level at R1 stage, while the differences of V13, R3 and R5 were not significant. There was no significant difference in root surface area between the two cultivars at the interaction stage of R3 and R5. There was no significant difference in the total root surface area at V13 stage under different nitrogen levels, and the difference was extremely significant ($p < 0.01$) at R1, R3 and R5 stages. With the increase of nitrogen levels, the total root surface area increased significantly. For JD27, the total root surface area of 150N was higher than that of 000N. For SD19, the total surface area of root increased by 29.32% and 62.44% respectively at 150N and 300N compared with 000N. Under 000N and 075N, the change of root pixel area of JD27 was significantly higher than that of SD19, but under 225N and 300N, the change of root pixel area of SD19 was significantly higher than that of JD27, which indicated that JD27 was more conducive to the

increase of root total surface area than SD19 under low nitrogen level, and high nitrogen level was more conducive to the increase of root total surface area of SD19 (Table 6).

**Table 6.** Effects of nitrogen levels on root surface area of maize varieties.

| Varieties | Nitrogen Levels | V13 (cm² plant⁻¹) | R1 (cm² plant⁻¹) | R3 (cm² plant⁻¹) | R5 (cm² plant⁻¹) |
|---|---|---|---|---|---|
| JD27 | 000N | 1303.43 abc | 1367.97 cd | 1850.92 cde | 1698.88 c |
| JD27 | 075N | 1308.13 abc | 1706.35 bc | 2183.75 bcd | 1787.22 bc |
| JD27 | 150N | 1285.06 abc | 1292.26 d | 1933.52 cde | 1787.13 bc |
| JD27 | 225N | 1157.30 bc | 1379.05 cd | 2482.08 b | 1594.61 cd |
| JD27 | 300N | 1287.58 abc | 1415.88 cd | 2250.25 bcd | 1775.34 c |
| SD19 | 000N | 995.82 c | 1219.36 d | 1582.49 e | 1216.59 d |
| SD19 | 075N | 1196.39 bc | 1297.92 cd | 1813.65 de | 1417.96 cd |
| SD19 | 150N | 1414.72 abc | 1581.73 bcd | 1961.19 bcde | 1526.65 cd |
| SD19 | 225N | 1822.59 a | 2490.35 a | 3052.28 a | 2280.09 a |
| SD19 | 300N | 1630.22 ab | 1897.38 b | 2347.02 bc | 2270.72 ab |
| F value | Varieties (V) | 1.65 | 10.99 ** | 0.01 | 0.02 |
| F value | Nitrogen (N) | 1.29 | 7.48 ** | 12.37 ** | 5.41 ** |
| F value | V:N | 2.32 | 10.86 ** | 2.57 | 6.94 ** |
| JD27 | | 1268.30 a | 1432.30 b | 2140.10 a | 1728.64 a |
| SD19 | | 1411.95 a | 1697.35 a | 2151.33 a | 1742.40 a |
| | 000N | 1149.63 a | 1293.67 c | 1716.70 c | 1457.74 c |
| | 075N | 1252.26 a | 1502.14 bc | 1998.70 bc | 1602.59 c |
| | 150N | 1349.89 a | 1436.99 bc | 1947.35 bc | 1656.89 bc |
| | 225N | 1489.94 a | 1934.70 a | 2767.18 a | 1937.35 ab |
| | 300N | 1458.90 a | 1656.63 b | 2298.63 b | 2023.03 a |

Different lowercase letters are significant difference at $p < 5\%$ on treatment, ** $p < 0.01$.

### 3.7. Total Root Volume

Factor analysis of variance showed that there was no significant difference in root volume between JD27 and SD19 at R3 and R5, but the difference was extremely significant ($p < 0.05$) at V13 and R1. Under the interaction of varieties and nitrogen levels, the total root volume reached a significant difference ($p < 0.05$) at V13, R3 and R5 stages, and reached a very significant difference ($p < 0.05$) at R1 stage. There was no significant difference in the total root volume at V13 stage under different nitrogen levels, but the difference was significant ($p < 0.05$) at R5 stage, and extremely significant ($p < 0.05$) at R1 and R3 stages. With the increase of nitrogen levels, the total root volume increased significantly, but the difference was different for different varieties. For JD27, 150N was higher than that of 000N. For SD19, the total root volume of 150N increased by 41.97% and 300N increased by 87.08%. The variation of total root volume of JD27 was slightly higher than that of SD19 under 000N and 075N. The variation of total root volume of SD19 was significantly higher than that of JD27 under 225N and 300N. The variation of total root volume of JD27 among nitrogen levels was not significant, indicating that it was less affected by nitrogen levels, and high nitrogen level was more conducive to the increase of total root volume of SD19 (Table 7).

### 3.8. Logistic Fitting of Plant Dry Matter Accumulation

The difference of dry matter accumulation between JD27 and SD19 was significant ($p < 0.05$). The dry matter accumulation of JD27 in V7, V13, R1, R3 and R5 was 6.25%, 5.65%, 10.06%, 5.11% and 4.88% higher than that of SD19, respectively. However, the difference of dry matter accumulation between JD27 and SD19 was not significant in V13, and the dry matter accumulation of JD27 did not reach significant level under the interaction of variety and nitrogen levels. With the increase of nitrogen levels, the dry matter accumulation of varieties also increased, but the difference was different for different varieties. For JD27, the total dry matter accumulation of 150N increased by 25.55% compared with 000N at each growth stage, and that of 300N increased by 25.55% compared with 000N. For SD19,

the total dry matter of 150N was 40.54% higher than that of 000N, and the total dry matter of 300N was 62.73% higher than that of 000N. The dry matter accumulation of SD19 was significantly lower than that of JD27, indicating that nitrogen levels were more conducive to the improvement of dry matter quality of SD19. The results showed that the difference of dry matter quality between the two varieties increased first, then decreased and then increased with the increase of nitrogen levels. The difference was the largest under 000N, followed by 300N, and the smallest under 150N. The results showed that JD27 was better than SD19 in increasing dry matter under low nitrogen levels, and high nitrogen level was more conducive to improving the dry matter quality of SD19 (Tables 8 and 9).

**Table 7.** Effects of different nitrogen levels on root overall volume of maize varieties.

| Varieties | Nitrogen Levels | V13 (cm³ plant⁻¹) | R1 (cm³ plant⁻¹) | R3 (cm³ plant⁻¹) | R5 (cm³ plant⁻¹) |
|---|---|---|---|---|---|
| JD27 | 000N | 97.69 abc | 98.63 cd | 122.78 cd | 110.45 bcd |
| JD27 | 075N | 90.59 bc | 101.12 cd | 170.66 bcd | 131.12 bcd |
| JD27 | 150N | 74.47 c | 75.94 d | 179.42 bcd | 130.41 bcd |
| JD27 | 225N | 61.66 c | 97.53 cd | 212.72 b | 117.14 bcd |
| JD27 | 300N | 87.07 c | 97.06 cd | 185.05 bc | 138.11 abc |
| SD19 | 000N | 75.87 c | 92.82 cd | 136.68 cd | 77.42 d |
| SD19 | 075N | 74.56 c | 86.58 d | 110.73 d | 98.61 cd |
| SD19 | 150N | 122.56 abc | 143.17 bc | 155.11 bcd | 122.62 bcd |
| SD19 | 225N | 158.05 ab | 224.97 a | 300.25 a | 199.61 a |
| SD19 | 300N | 166.83 a | 161.77 b | 214.61 b | 172.93 ab |
| F value | Varieties (V) | 8.25 ** | 18.02 ** | 0.48 | 0.47 |
| F value | Nitrogen (N) | 1.54 | 4.97 ** | 11.46 ** | 3.64 * |
| F value | V:N | 3.49 * | 5.43 ** | 3.42 * | 2.99 * |
| JD27 | | 82.29 b | 94.06 b | 174.13 a | 125.45 a |
| SD19 | | 119.57 a | 141.86 a | 183.48 a | 134.24 a |
| | 000N | 86.78 a | 95.73 b | 129.73 c | 93.94 b |
| | 075N | 82.58 a | 93.85 b | 140.70 c | 114.87 ab |
| | 150N | 98.51 a | 109.55 b | 167.26 bc | 126.52 ab |
| | 225N | 109.85 a | 161.25 a | 256.49 a | 158.37 a |
| | 300N | 126.95 a | 129.41 ab | 199.83 b | 155.52 a |

Different lowercase letters are significant difference at $p < 5\%$ on treatment, ** $p < 0.01$, * $p < 0.05$.

**Table 8.** Dry matter accumulation of maize varieties with different nitrogen levels at different growth stages.

| Varieties | Nitrogen Levels | V7 (t ha⁻¹) | V13 (t ha⁻¹) | R1 (t ha⁻¹) | R3 (t ha⁻¹) | R5 (t ha⁻¹) |
|---|---|---|---|---|---|---|
| JD27 | 000N | 1.71 d | 3.80 c | 7.48 fg | 13.99 f | 17.88 f |
| JD27 | 075N | 2.32 bc | 5.24 ab | 8.50 def | 16.51 e | 19.86 de |
| JD27 | 150N | 2.49 bc | 5.54 a | 9.32 cde | 17.62 cde | 21.35 bcd |
| JD27 | 225N | 2.60 ab | 5.70 a | 10.38 c | 18.95 abc | 22.55 abc |
| JD27 | 300N | 2.80 a | 5.92 a | 13.58 a | 20.43 a | 23.73 a |
| SD19 | 000N | 1.40 d | 3.45 c | 6.28 g | 12.04 g | 15.63 g |
| SD19 | 075N | 2.27 c | 4.44 bc | 8.15 ef | 15.84 ef | 19.08 ef |
| SD19 | 150N | 2.37 bc | 5.48 a | 8.77 de | 17.04 de | 20.87 cd |
| SD19 | 225N | 2.54 abc | 5.61 a | 9.75 cd | 18.53 bcd | 21.80 bc |
| SD19 | 300N | 2.64 ab | 5.81 a | 11.82 b | 19.80 ab | 23.07 ab |
| F value | Varieties (V) | 4.83 * | 1.76 | 14.05 ** | 5.46 * | 7.79 * |
| F value | Nitrogen (N) | 41.23 ** | 14.66 ** | 65.94 ** | 44.12 ** | 42.90 ** |
| F value | V:N | 0.58 | 0.43 | 1.14 | 0.57 | 0.83 |
| JD27 | | 2.38 a | 5.24 a | 9.85 a | 17.50 a | 21.07 a |
| SD19 | | 2.24 b | 4.96 a | 8.95 b | 16.65 b | 20.09 b |
| | 000N | 1.55 d | 3.63 c | 6.88 d | 13.01 d | 16.76 d |
| | 075N | 2.29 c | 4.84 b | 8.32 c | 16.18 c | 19.47 c |
| | 150N | 2.43 bc | 5.51 ab | 9.04 c | 17.33 c | 21.11 b |
| | 225N | 2.57 ab | 5.66 a | 10.06 b | 18.74 b | 22.18 b |
| | 300N | 2.72 a | 5.87 a | 12.70 a | 20.11 a | 23.40 a |

Different lowercase letters are significant difference at $p < 5\%$ on treatment, ** $p < 0.01$, * $p < 0.05$.

**Table 9.** Logistic fitting and parameter of maize dry matter under nitrogen levels.

| Treatment | Fitting Equation | $R^2$ | $V$max ($kg\ ha^{-1}\ d^{-1}$) | $V$a ($kg\ ha^{-1}\ d^{-1}$) | $T_m$ (d) | $T_{2-1}$ (d) |
|---|---|---|---|---|---|---|
| JD27-000N | y = 19.07/(1 + 219.67exp(−0.06x)) | 0.968 | 300.94 | 147.79 | 85.40 | 41.50 |
| JD27-075N | y = 21.26/(1 + 168.77exp(−0.06x)) | 0.993 | 327.80 | 164.84 | 83.17 | 42.49 |
| JD27-150N | y = 22.81/(1 + 166.94exp(−0.06x)) | 0.996 | 351.37 | 176.84 | 83.06 | 42.53 |
| JD27-225N | y = 23.77/(1 + 201.99exp(−0.07x)) | 0.992 | 386.64 | 184.30 | 81.60 | 40.28 |
| JD27-300N | y = 24.03/(1 + 315.45exp(−0.08x)) | 0.995 | 452.09 | 186.27 | 76.45 | 34.81 |
| Average | | | 363.77 | 172.01 | 81.94 | 40.32 |
| SD19-000N | y = 16.84/(1 + 199.99exp(−0.06x)) | 0.961 | 257.67 | 130.51 | 86.54 | 42.79 |
| SD19-075N | y = 20.29/(1 + 219.57exp(−0.06x)) | 0.978 | 327.94 | 157.31 | 83.41 | 40.54 |
| SD19-150N | y = 22.48/(1 + 156.84exp(−0.06x)) | 0.985 | 337.92 | 174.26 | 84.07 | 43.57 |
| SD19-225N | y = 23.08/(1 + 205.97exp(−0.07x)) | 0.995 | 375.65 | 178.92 | 81.84 | 40.24 |
| SD19-300N | y = 23.84/(1 + 251.83exp(−0.07x)) | 0.997 | 417.33 | 184.84 | 78.97 | 37.42 |
| Average | y = 19.07/(1 + 219.67exp(−0.06x)) | | 343.30 | 165.17 | 82.97 | 40.91 |

Note: y = a/(1 + b*exp(−k*x)), *V*max: Maximum growth rate; *V*a: Average growth rate; *T*m: Time of instantaneous maximum slope; $T_{2-1}$:Rapid growth rate period.

Figure 2 showed that the distribution of dry matter quality in the whole growth period of maize was an "S" type growth curve. The ten curves fitted in this experiment all reached significant ($p < 0.05$) difference level (R2 > 0.96). The effects of nitrogen levels and different maize varieties on the maximum theoretical biomass (a), maximum accumulation rate (Vmax), average accumulation rate (VA) and maximum accumulation rate of maize were significant with the increase of nitrogen levels, a, Vmax and VA increased, while TM and $T_{2-1}$ decreased. The dry matter accumulation rate of JD27 was 363.77 kg ha$^{-1}$ d$^{-1}$ on 81.94 days after sowing, and that of SD19 was 343.30 kg ha$^{-1}$ d$^{-1}$ on 82.97 days after sowing. The Vmax and VA of JD27 were higher than those of SD19 at the time when the maximum accumulation rate appeared, indicating that the dry matter absorption efficiency of JD27 was better than that of SD19.

*3.9. Plant Dry Matter Accumulation*

With the development of maize, the dry matter accumulation of two maize varieties at different growth stages showed a trend of first increasing and then decreasing. JD27 and SD19 had the most significant dry matter accumulation at R1-R3 stage, which were 7.65 t ha$^{-1}$ and 7.7 t ha$^{-1}$, respectively, but the difference between the two varieties did not reach a significant level. Except in R1-R3 stage, the dry matter accumulation of JD27 was higher than that of SD19, and it was 6.25%, 5.54%, 15.54% and 3.78% higher in emergence-V7, V7-V13, V13-R1 and R3-R5, respectively. At sowing-V7 stage, the dry matter accumulation of JD27 was significantly higher than that of SD19 ($p < 0.05$), but the difference was not significant at other stages, which indicated that JD27 had higher nitrogen use efficiency at the early stage of growth and development. The difference of dry matter accumulation between the two varieties decreased with the increase of nitrogen levels. The total dry matter weight of JD27 was 14.40%, 4.09%, 2.30%, 3.53% and 2.91% higher than that of SD19 at 000N, 075N, 150N, 225N and 300N levels, respectively. The dry matter accumulation of maize reached significant ($p < 0.05$) or extremely significant ($p < 0.05$) difference at emergence-V7, V7-V13, V13-R1 and R1-R3 growth stages, but there was no significant difference at R3-R5 growth stage, which indicated that JD27 with low nitrogen efficiency had stronger ability of nitrogen absorption and biomass synthesis and could make full use of nitrogen at the later stage of maize growth and development (Table 10).

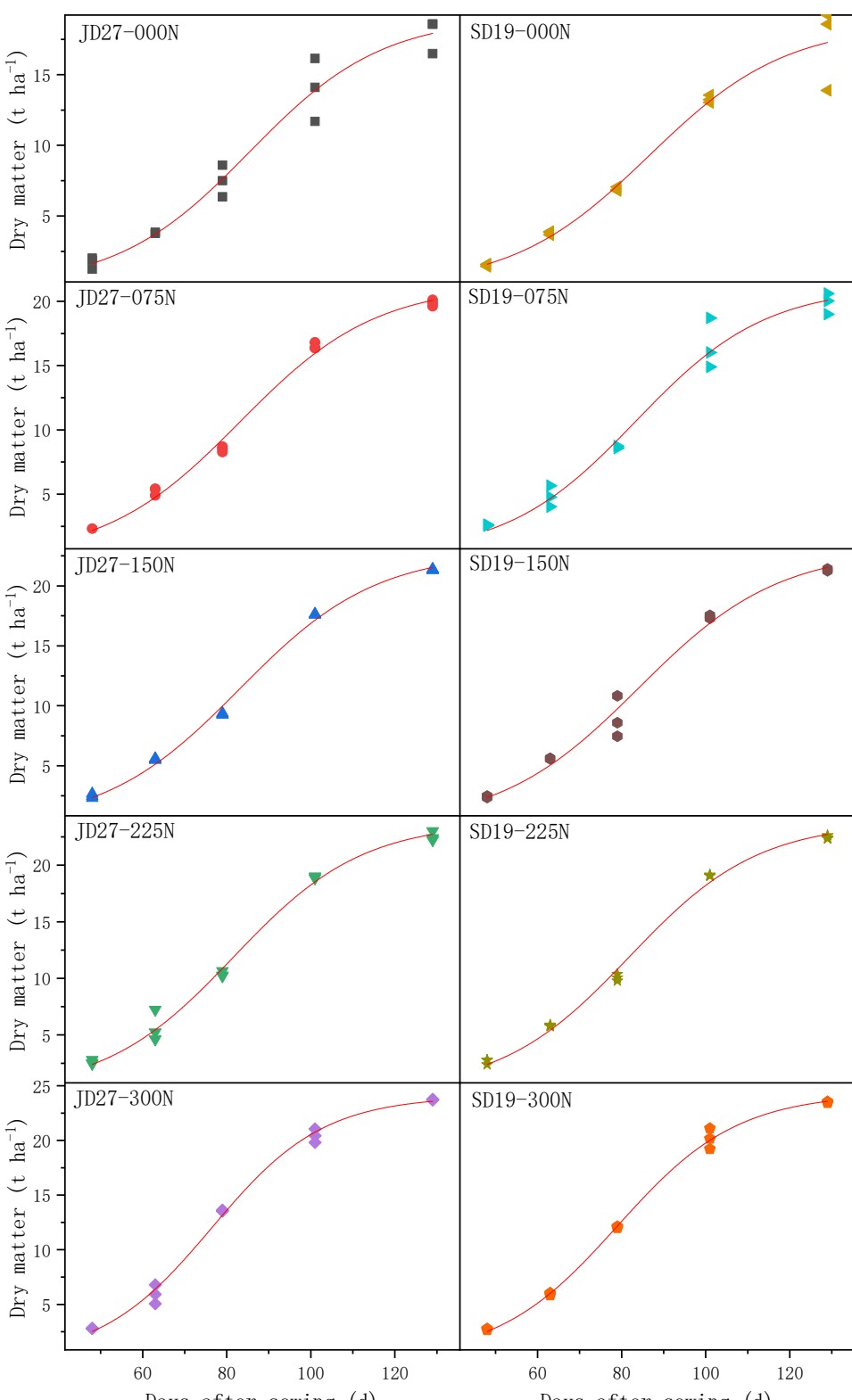

**Figure 2.** Logistic fitting curves of JD27 and SD19 with different nitrogen levels.

**Table 10.** Dry matter accumulation of maize varieties in growth periods with different nitrogen levels.

| Varieties | Nitrogen Levels | Emergence -V7 (t ha$^{-1}$) | V7–V13 (t ha$^{-1}$) | V13–R1 (t ha$^{-1}$) | R1–R3 (t ha$^{-1}$) | R3–R5 (t ha$^{-1}$) |
|---|---|---|---|---|---|---|
| JD27 | 000N | 1.71 d | 2.09 a | 3.68 cd | 6.50 ab | 3.90 a |
| JD27 | 075N | 2.32 bc | 2.92 a | 3.26 cd | 8.01 ab | 3.35 a |
| JD27 | 150N | 2.49 bc | 3.05 a | 3.78 cd | 8.30 a | 3.73 a |
| JD27 | 225N | 2.60 ab | 3.10 a | 4.68 bc | 8.58 a | 3.60 a |
| JD27 | 300N | 2.80 a | 3.12 a | 7.65 a | 6.86 ab | 3.30 a |
| SD19 | 000N | 1.40 d | 2.06 a | 2.82 d | 5.77 b | 3.58 a |
| SD19 | 075N | 2.27 c | 2.17 a | 3.71 cd | 7.69 ab | 3.24 a |
| SD19 | 150N | 2.37 bc | 3.10 a | 3.29 cd | 8.27 a | 3.84 a |
| SD19 | 225N | 2.54 abc | 3.07 a | 4.13 cd | 8.78 a | 3.27 a |
| SD19 | 300N | 2.64 ab | 3.17 a | 6.01 ab | 7.97 ab | 3.27 a |
| F value | Varieties (V) | 4.83 * | 0.36 | 3.57 | 0.01 | 0.06 |
| F value | Nitrogen (N) | 41.23 ** | 3.13 * | 16.44 ** | 3.46 * | 0.15 |
| F value | V:N | 0.58 | 0.42 | 1.06 | 0.43 | 0.02 |
| JD27 | | 2.38 a | 2.86 a | 4.61 a | 7.65 a | 3.57 a |
| SD19 | | 2.24 b | 2.71 a | 3.99 a | 7.70 a | 3.44 a |
| | 000N | 1.55 d | 2.07 b | 3.25 b | 6.13 b | 3.74 a |
| | 075N | 2.29 c | 2.54 ab | 3.48 b | 7.85 a | 3.29 a |
| | 150N | 2.43 bc | 3.08 a | 3.53 b | 8.29 a | 3.78 a |
| | 225N | 2.57 ab | 3.09 a | 4.40 b | 8.68 a | 3.44 a |
| | 300N | 2.72 a | 3.14 a | 6.83 a | 7.41 ab | 3.28 a |

Different lowercase letters are significant difference at $p < 5\%$ on treatment, ** $p < 0.01$, * $p < 0.05$.

### 3.10. Plant Dry Matter Accumulation Rate

Table 11 showed that JD27 and SD19 had the highest dry matter accumulation rate at R1-R3 stage, which were 347.73 kg ha$^{-1}$ d$^{-1}$ and 349.84 kg ha$^{-1}$ d$^{-1}$, respectively, but the difference between the two varieties did not reach a significant level. Except in R1-R3 stage, the dry matter accumulation rate of JD27 was higher than that of SD19, and it was 6.20%, 24.00%, 15.47% and 3.81% higher in emergence-V7, V7–13, V13–R1 and R3–R5, respectively. The difference of dry matter accumulation rate between the two varieties decreased with the increase of nitrogen levels. The total dry matter weight of JD27 was 14.60%, 4.63%, 2.80%, 3.81% and 4.72% higher than that of SD19 at 000N, 075N, 150N, 225N and 300N levels, respectively. The dry matter accumulation rate of maize reached significant ($p < 0.05$) or extremely significant ($p < 0.05$) difference at emergence-V7, V7-V13, V13–R1 and R1–R3 growth stages, but there was no significant difference at R3-R5 growth stage, which indicated that JD27, a low nitrogen efficient variety, had stronger nitrogen absorption efficiency and biomass synthesis ability at early growth stage (Table 11).

### 3.11. Dry Matter Distribution Ratio of Stem and Leaf Sheath

The dry matter distribution ratio of stem and leaf sheath increased first and then decreased with the growth and development of maize. The dry matter distribution ratio of stem and leaf sheath of SD19 was higher than that of JD27 at all stages except V13 stage, which indicated that JD27 accumulated faster dry matter in stem and leaf sheath at V13 stage. The dry matter accumulation ratio of stem and leaf sheath was significantly different in R1, R3 and R5 ($p < 0.05$). The dry matter accumulation ratio of stem and leaf sheath in 075N treatment was significantly higher than that in 225N treatment at R1 ($p < 0.05$), which indicated that low nitrogen level was not conducive to increase the total dry matter weight, while medium and high nitrogen levels could meet the needs of stem growth and development Please. At R3 stage, 000N treatment was significantly higher than 225N and 300N treatment ($p < 0.05$). At this stage, grain filling was just beginning, and a large amount of nitrogen and nutrition were urgently needed to transform to vegetative organs. At R5 stage, 000N treatment was significantly higher than other treatments ($p < 0.05$), indicating that dry matter increased nitrogen synthesis mainly transformed to reproductive organs at this stage, and dry matter synthesized at low nitrogen level could not meet the needs

of vegetative organ growth and development. Therefore, in the later stage of growth and development, with a large amount of dry matter transformed to vegetative organs, the accumulation of stem dry matter no longer increased or even lost, resulting in a gradual decrease in the proportion of dry matter (Table 12).

**Table 11.** Dry matter accumulation rate of maize varieties in growth periods with different nitrogen levels.

| Varieties | Nitrogen Levels | Sowing-V7 (kg ha$^{-1}$ d$-1$) | V7–V13 (kg ha$^{-1}$ d$-1$) | V13–R1 (kg ha$^{-1}$ d$-1$) | R1–R3 (kg ha$^{-1}$ d$-1$) | R3–R5 (kg ha$^{-1}$ d$-1$) |
|---|---|---|---|---|---|---|
| JD27 | 000N | 35.64 d | 139.31 a | 230.24 cd | 295.56 ab | 139.13 a |
| JD27 | 075N | 48.36 bc | 194.60 a | 203.75 cd | 364.18 ab | 119.49 a |
| JD27 | 150N | 51.80 bc | 203.63 a | 236.10 cd | 377.45 a | 133.17 a |
| JD27 | 225N | 54.08 ab | 206.93 a | 292.34 bc | 389.78 a | 128.51 a |
| JD27 | 300N | 58.40 a | 207.92 a | 478.39 a | 311.67 ab | 117.72 a |
| SD19 | 000N | 29.08 d | 137.20 a | 176.41 d | 262.13 b | 128.03 a |
| SD19 | 075N | 47.24 c | 144.69 a | 231.86 cd | 349.70 ab | 115.74 a |
| SD19 | 150N | 49.47 bc | 206.97 a | 205.44 cd | 375.89 a | 137.07 a |
| SD19 | 225N | 52.93 abc | 204.91 a | 258.25 cd | 399.20 a | 116.97 a |
| SD19 | 300N | 55.08 ab | 211.16 a | 375.84 ab | 362.31 ab | 116.79 a |
| F value | Varieties (V) | 4.83 * | 0.36 | 3.57 | 0.01 | 0.06 |
| F value | Nitrogen (N) | 41.23 ** | 3.13 * | 16.44 ** | 3.46 * | 0.15 |
| F value | V:N | 0.58 | 0.42 | 1.06 | 0.43 | 0.02 |
| JD27 | | 49.66 a | 190.48 a | 288.16 a | 347.73 a | 127.60 a |
| SD19 | | 46.76 b | 180.99 a | 249.56 a | 349.84 a | 122.92 a |
| | 000N | 32.36 d | 138.25 b | 203.33 b | 278.85 b | 133.58 a |
| | 075N | 47.80 c | 169.65 ab | 217.81 b | 356.94 a | 117.61 a |
| | 150N | 50.63 bc | 205.30 a | 220.77 b | 376.67 a | 135.12 a |
| | 225N | 53.51 ab | 205.92 a | 275.30 b | 394.49 a | 122.74 a |
| | 300N | 56.74 a | 209.54 a | 427.12 a | 336.99 ab | 117.25 a |

Different lowercase letters are significant difference at $p < 5\%$ on treatment, ** $p < 0.01$, * $p < 0.05$.

**Table 12.** Effects of nitrogen levels on dry matter distribution in maize stalk and leaf sheaths.

| Varieties | Nitrogen Levels | Ratio of Stalk and Leaf Sheaths (%) | | | | |
|---|---|---|---|---|---|---|
| | | V7 | V13 | R1 | R3 | R5 |
| JD27 | 000N | 39.46 a | 47.47 a | 58.97 abc | 31.79 bc | 25.38 ab |
| JD27 | 075N | 37.65 a | 49.13 a | 62.71 a | 27.46 cde | 23.26 bcde |
| JD27 | 150N | 37.22 a | 51.03 a | 55.63 bc | 24.5 de | 19.73 e |
| JD27 | 225N | 36.27 a | 58.67 a | 53.3 c | 23.44 e | 20.39 de |
| JD27 | 300N | 41.18 a | 54.09 a | 55.25 bc | 24.43 de | 20.94 cde |
| SD19 | 000N | 36.78 a | 48.87 a | 60.93 ab | 39.68 a | 28.74 a |
| SD19 | 075N | 41.36 a | 44.04 a | 60.84 ab | 34.45 ab | 24.16 bcd |
| SD19 | 150N | 37.43 a | 53.33 a | 56.01 bc | 30 bcd | 24.45 bc |
| SD19 | 225N | 40.27 a | 44.41 a | 55.89 bc | 29.15 bcde | 23.39 bcde |
| SD19 | 300N | 38.72 a | 49.8 a | 58.81 abc | 28.5 bcde | 22.07 bcde |
| F value | Varieties (V) | 0.17 | 1.65 | 1.14 | 24.74 ** | 11.92 ** |
| F value | Nitrogen (N) | 0.5 | 0.54 | 4.6 ** | 8.9 ** | 7.26 ** |
| F value | V:N | 1.13 | 0.91 | 0.59 | 0.29 | 0.89 |
| JD27 | | 38.36 a | 52.08 a | 57.17 a | 26.33 b | 21.94 b |
| SD19 | | 38.91 a | 48.09 a | 58.49 a | 32.36 a | 24.56 a |
| | 000N | 38.12 a | 48.17 a | 59.95 ab | 35.74 a | 27.06 a |
| | 075N | 39.51 a | 46.59 a | 61.78 a | 30.96 b | 23.71 b |
| | 150N | 37.33 a | 52.18 a | 55.82 bc | 27.25 bc | 22.09 b |
| | 225N | 38.27 a | 51.54 a | 54.59 c | 26.3 c | 21.89 b |
| | 300N | 39.95 a | 51.95 a | 57.03 bc | 26.47 c | 21.51 b |

Different lowercase letters are significant difference at $p < 5\%$ on treatment, ** $p < 0.01$.

### 3.12. Distribution Ratio of Dry Matter in Leaves

The results showed that the dry matter distribution ratio of leaves decreased gradually with the growth and development of maize. The difference in dry matter distribution ratio of leaves between the two varieties reached a significant level ($p < 0.05$) at R1 stage. The dry matter ratio of leaves of JD27 and SD19 were 6.26% and 5.51% respectively. The dry matter distribution ratio of JD27 was higher than that of SD19 at all growth stages, indicating that the function of JD27 was better than that of SD19. The results showed that nitrogen levels affected the dry matter accumulation ratio of maize leaves at different stages, and the difference was extremely significant ($p < 0.05$) at R1 stage. Under 225N treatment at R1 stage, the dry matter accumulation ratio of maize leaves was significantly higher ($p < 0.05$) than that of low nitrogen levels, indicating that low nitrogen level was not conducive to leaf growth and dry matter accumulation, while medium and high nitrogen levels promoted the rapid growth and dry matter accumulation of maize leaves nitrogen level can meet the needs of leaf growth and development (Table 13).

**Table 13.** Effects of nitrogen levels on dry matter distribution in maize leaf blades.

| Varieties | Nitrogen Levels | Ratio of Leaf Blades (%) | | | | |
|---|---|---|---|---|---|---|
| | | V7 | V13 | R1 | R3 | R5 |
| JD27 | 000N | 60.54 a | 52.53 a | 33.43 abcd | 18.85 a | 15.29 ab |
| JD27 | 075N | 62.35 a | 50.87 a | 29.89 bcd | 16.98 a | 13.86 b |
| JD27 | 150N | 62.78 a | 48.97 a | 36.4 a | 17.83 a | 17.65 a |
| JD27 | 225N | 63.73 a | 41.33 a | 33.99 abc | 18.44 a | 14.27 b |
| JD27 | 300N | 58.82 a | 45.91 a | 35.14 ab | 17.42 a | 13.71 b |
| SD19 | 000N | 63.22 a | 51.13 a | 29.37 cd | 18.5 a | 13.94 b |
| SD19 | 075N | 58.64 a | 55.96 a | 28.06 d | 16.63 a | 13.04 b |
| SD19 | 150N | 62.57 a | 46.67 a | 33.78 abc | 18.2 a | 13.64 b |
| SD19 | 225N | 59.73 a | 55.59 a | 34.14 abc | 16.84 a | 13.65 b |
| SD19 | 300N | 61.28 a | 50.2 a | 31.68 abcd | 18.99 a | 14.66 ab |
| F value | Varieties (V) | 0.17 | 1.65 | 4.93 * | 0.01 | 3.8 |
| F value | Nitrogen (N) | 0.5 | 0.54 | 4.17 * | 0.93 | 1.51 |
| F value | V:N | 1.13 | 0.91 | 0.48 | 0.64 | 1.81 |
| JD27 | | 61.64 a | 47.92 a | 33.77 a | 17.9 a | 14.95 a |
| SD19 | | 61.09 a | 51.91 a | 31.41 b | 17.83 a | 13.78 a |
| | 000N | 61.88 a | 51.83 a | 31.4 bc | 18.68 a | 14.61 ab |
| | 075N | 60.49 a | 53.41 a | 28.97 c | 16.8 a | 13.45 b |
| | 150N | 62.67 a | 47.82 a | 35.09 a | 18.01 a | 15.64 a |
| | 225N | 61.73 a | 48.46 a | 34.06 ab | 17.64 a | 13.96 ab |
| | 300N | 60.05 a | 48.05 a | 33.41 ab | 18.2 a | 14.19 ab |

Different lowercase letters are significant difference at $p < 5\%$ on treatment, * $p < 0.05$.

### 3.13. Dry Matter Distribution Ratio of Tassel, Petiole Bract Axis and Grain

The dry matter allocation ratio of male ear of both varieties reached the maximum at R1 and the minimum at R5, and the difference of dry matter allocation ratio of male ear of both varieties reached extremely significant ($p < 0.05$) level at R1 and R3. The dry matter distribution ratio of male panicle of JD27 was lower than that of SD19 at all growth stages, indicating that dry matter accumulation of male panicle of JD27 was less than that of SD19. The dry matter distribution ratio of petiole bract axis increased gradually with the growth and development of maize. Both varieties reached the maximum at R5 stage and the minimum at R1 stage. The difference of dry matter distribution ratio of petiole bract axis between the two varieties reached extremely significant ($p < 0.05$) and significant ($p < 0.05$) levels at R1 and R5 stages. The dry matter distribution ratio of petiole bract axis of JD27 was higher than that of SD19 at all growth stages, indicating that the dry matter accumulation of petiole bract axis of JD27 was higher than that of SD19. At R1 stage, the dry matter accumulation ratio of petiole bract axis in 225N treatment was significantly higher than that in low nitrogen levels ($p < 0.05$), indicating that low nitrogen level was not

conducive to the growth and dry matter accumulation of petiole bract axis, while medium and high nitrogen levels promoted the rapid growth and dry matter accumulation of petiole bract axis. Grain dry matter distribution ratio increased gradually with the growth and development of maize. The two varieties reached the maximum at R5 stage and the minimum at R3 stage. The difference of grain dry matter distribution ratio between the two varieties reached a very significant ($p < 0.05$) level at R3 stage. During the grain filling period, the dry matter distribution ratio of JD27 was higher than that of SD19, indicating that the dry matter accumulation of JD27 was higher than that of SD19. At R3 stage, the proportion of grain dry matter accumulation under 225N treatment was significantly higher than that under low nitrogen levels ($p < 0.05$), indicating that low nitrogen level was not conducive to grain growth and dry matter accumulation, while medium and high nitrogen levels promoted rapid grain growth and dry matter accumulation. At R5 stage, the grain dry matter distribution ratio of nitrogen levels was significantly higher than that of nitrogen deficiency treatment ($p < 0.05$), which further indicated that nitrogen fertilizer promoted the increase of grain dry matter (Table 14).

**Table 14.** Effects of nitrogen levels on dry matter distribution in tassel, shark-husk leaves-cob and grain.

| Varieties | Nitrogen Levels | Ratio of Tassel (%) | | | Ratio of Shark-Husk Leaves-Cob (%) | | | Grain (%) | |
|---|---|---|---|---|---|---|---|---|---|
| | | R1 | R3 | R5 | R1 | R3 | R5 | R3 | R5 |
| JD27 | 000N | 2.78 bc | 1.06 bc | 0.99 abc | 4.83 bc | 13.85 a | 11.14 ab | 34.44 d | 47.2 bc |
| JD27 | 075N | 2.45 bc | 0.89 c | 0.63 bc | 4.95 bc | 14.25 a | 10.76 ab | 40.41 abc | 51.49 a |
| JD27 | 150N | 1.94 c | 0.66 c | 0.53 bc | 6.03 bc | 13.7 a | 12.21 a | 43.32 ab | 49.89 abc |
| JD27 | 225N | 2.6 bc | 0.9 c | 1.15 abc | 10.11 a | 13.08 a | 11.81 ab | 44.14 a | 52.38 a |
| JD27 | 300N | 2.4 bc | 0.82 c | 0.75 bc | 7.21 ab | 15.56 a | 11.53 ab | 41.78 abc | 53.08 a |
| SD19 | 000N | 6.42 a | 2.37 a | 1.7 a | 3.28 c | 12.6 a | 9.62 b | 26.85 e | 46.00 c |
| SD19 | 075N | 5.98 a | 1.9 ab | 1.36 ab | 5.12 bc | 12.89 a | 10.63 ab | 34.13 d | 50.82 ab |
| SD19 | 150N | 6.36 a | 2.17 a | 0.78 bc | 3.84 bc | 12.88 a | 10.47 ab | 36.76 cd | 50.67 ab |
| SD19 | 225N | 3.54 b | 1.59 ab | 0.92 abc | 6.44 bc | 13.96 a | 10.73 ab | 38.46 bc | 51.31 ab |
| SD19 | 300N | 5.64 a | 1.11 bc | 0.5 c | 3.87 bc | 12.61 a | 10.72 ab | 38.79 abc | 52.04 a |
| F value | Varieties (V) | 121.53 ** | 24.63 ** | 2.16 | 9.5 ** | 2.44 | 4.83 * | 31.93 ** | 0.54 |
| F value | Nitrogen (N) | 3.17 * | 1.61 | 2.66 | 4.38 * | 0.18 | 0.58 | 14.1 ** | 5.75 ** |
| F value | V:N | 4.22 * | 1.27 | 1.7 | 1.01 | 0.76 | 0.35 | 0.56 | 0.18 |
| JD27 | | 2.44 b | 0.87 b | 0.81 a | 6.62 a | 14.09 a | 11.49 a | 40.82 a | 50.81 a |
| SD19 | | 5.59 a | 1.83 a | 1.05 a | 4.51 b | 10.99 a | 12.43 b | 35 b | 50.17 a |
| | 000N | 4.6 a | 1.72 a | 1.35 a | 4.05 b | 13.22 a | 10.38 a | 30.64 c | 46.6 b |
| | 075N | 4.22 a | 1.4 ab | 1 ab | 5.03 b | 13.57 a | 10.69 a | 37.27 b | 51.15 a |
| | 150N | 4.15 a | 1.41 ab | 0.65 b | 4.94 b | 13.29 a | 11.34 a | 40.04 ab | 50.28 a |
| | 225N | 3.07 b | 1.24 ab | 1.03 ab | 8.28 a | 13.52 a | 11.27 a | 41.3 a | 51.85 a |
| | 300N | 4.02 a | 0.96 b | 0.62 b | 5.54 b | 14.08 a | 11.12 a | 40.29 ab | 52.56 a |

Different lowercase letters are significant difference at $p < 5\%$ on treatment, ** $p < 0.01$, * $p < 0.05$.

### 3.14. Dry Matter Transport

In terms of dry matter assimilation after anthesis, the APA and CAPA values of JD27 were higher than those of SD19, and the higher ratios were 9.29% and 12.17%, respectively. It can be seen that the pre anthesis transport capacity of SD19 is better than that of JD27, and the post anthesis dry matter assimilation capacity of JD27 is outstanding. The higher post anthesis dry matter assimilation capacity of JD27 inhibits the pre anthesis dry matter transport, and its APA mainly comes from the post anthesis dry matter assimilation effect. Under different nitrogen levels, there were significant differences in APA nitrogen levels ($p < 0.05$), which showed that the middle and high nitrogen levels was better than the nitrogen deficiency treatment, indicating that nitrogen played a crucial role in the increase of dry matter assimilation after anthesis (Table 15).

**Table 15.** Effects of nitrogen levels on dry matter redistribution of pre-silking and transformation of post-silking in maize.

| Varieties | Nitrogen Levels | RPA (t ha$^{-1}$) | PRAP (%) | CRAP (%) | APA (t ha$^{-1}$) | CAPA (%) |
|---|---|---|---|---|---|---|
| JD27 | 000N | 1.37 a | 11.31 a | 13.68 a | 8.32 bc | 86.32 a |
| JD27 | 075N | 0.25 a | 2.28 a | 2.26 a | 9.60 abc | 97.74 a |
| JD27 | 150N | 0.67 a | 8.04 a | 5.42 a | 12.18 a | 94.58 a |
| JD27 | 225N | 0.40 a | 4.52 a | 3.09 a | 12.54 a | 96.91 a |
| JD27 | 300N | 0.57 a | 7.38 a | 4.14 a | 10.93 ab | 95.86 a |
| SD19 | 000N | 2.23 a | 25.58 a | 26.49 a | 6.75 c | 73.51 a |
| SD19 | 075N | 1.26 a | 13.63 a | 12.40 a | 9.59 abc | 87.60 a |
| SD19 | 150N | 1.97 a | 22.53 a | 16.05 a | 9.95 abc | 83.95 a |
| SD19 | 225N | 1.74 a | 19.75 a | 13.61 a | 11.07 ab | 86.39 a |
| SD19 | 300N | 1.45 a | 18.38 a | 11.21 a | 11.65 ab | 88.79 a |
| F value | Varieties (V) | 3.78 | 3.84 | 3.36 | 1.81 | 3.36 |
| F value | Nitrogen (N) | 0.4 | 0.26 | 0.73 | 5.23 ** | 0.73 |
| F value | V:N | 0.03 | 0.02 | 0.03 | 0.65 | 0.03 |
| JD27 | | 0.65 a | 6.71 a | 5.72 a | 10.71 a | 94.28 a |
| SD19 | | 1.73 a | 19.98 a | 15.95 a | 9.80 a | 84.05 a |
| | 000N | 1.80 a | 18.45 a | 20.09 a | 7.54 b | 79.92 a |
| | 075N | 0.75 a | 7.95 a | 7.33 a | 9.60 ab | 92.67 a |
| | 150N | 1.32 a | 15.29 a | 10.73 a | 11.06 a | 89.27 a |
| | 225N | 1.07 a | 12.14 a | 8.35 a | 11.81 a | 91.65 a |
| | 300N | 1.01 a | 12.88 a | 7.67 a | 11.29 a | 92.33 a |

Different lowercase letters are significant difference at $p < 5\%$ on treatment, ** $p < 0.01$.

### 3.15. Maize Yield and Its Components

In terms of varieties, the ear length of SD19 was 19.16% and 12.72% higher than that of JD27 in two years ($p < 0.05$), and the ear diameter of JD27 was 6.77% higher than that of SD19 in 2016 ($p < 0.05$). There was no significant difference in 2015. The average number of grains per ear of the two varieties was basically the same, and there was no significant difference between the two varieties, with an average range of 519 grains to 596 grains. The moisture content of SD19 was slightly higher than that of JD27 in two years, which indicated that the dehydration rate of SD19 was slower than that of JD27 in the later growth stage, showing the phenomenon of late maturity. There was no significant difference in 100 grain weight between the two varieties, ranging from 33.22g to 34.84g. The yield of JD27 was 11.02% and 18.44% higher than that of SD19 in two years ($p < 0.05$), showing good high yield (Table 16).

With the increase of nitrogen levels, ear length and ear diameter showed an increasing trend, ear length with nitrogen levels in two years was significantly ($p < 0.05$) greater than that with 000N, ear diameter in 2016 with nitrogen levels was significantly ($p < 0.05$) greater than that of with 000N, there was no significant difference in 2015, ear length and ear diameter in 2015 were higher than that in 2016 as a whole. With the increase of nitrogen levels, the bald tip length decreased first and then increased. Both nitrogen deficiency and nitrogen enrichment could promote the bald tip length, which indicated that the appropriate amount of nitrogen levels was more conducive to promote the formation of full ear. The difference of grain number per spike in two years was significant ($p < 0.05$). In 2015, 300N was significantly ($p < 0.05$) greater than 000N. In 2016, fertilizer treatment was significantly ($p < 0.05$) greater than 000N. With the increase of nitrogen levels, the grain water content showed a decreasing trend, which indicated that increasing nitrogen levels were helpful to accelerate the dehydration of grain at the late growth stage. In the two years, the 100-grain weight of 150N, 225N and 300N was significantly higher than that of 000N and 075N ($p < 0.05$), which indicated that increasing nitrogen levels was beneficial to increase dry matter accumulation and plumpness of grains. With the increase of nitrogen levels, the yield increased first and then decreased slightly. The yield in 2015 was higher than that in 2016, and the ratio of increase to increase was 21.34%, 2.93%, 11.81%, 16.74% and 18.14% respectively. The yield of maize under 150N, 225N and 300N

was significantly higher than that of under 000N and 075N ($p < 0.05$), which indicated that increasing nitrogen levels was helpful to promote the formation of yield, but there was no significant difference in yield under medium and high nitrogen levels, which indicated that excessive fertilization could not significantly improve yield, but caused fertilizer waste, and a higher yield could be obtained under appropriate fertilizer amount (Table 16).

**Table 16.** Effects of nitrogen levels on maize yield and its components (2015 and 2016).

| Year | Varieties | Nitrogen Levels | Ear Length (cm) | Ear Diameter (cm) | Ear Tip Length (cm) | Grains Per Ear | Moisture (%) | 100-Kernel Weight (g) | Yield (t ha⁻¹) |
|---|---|---|---|---|---|---|---|---|---|
| 2015 | JD27 | 000N | 18.30 c | 4.97 a | 0.60 abc | 541.07 b | 26.37 d | 29.65 c | 6.34 de |
| | JD27 | 075N | 18.43 c | 5.17 a | 0.17 bc | 577.60 ab | 29.53 bc | 33.33 ab | 8.05 bc |
| | JD27 | 150N | 19.30 c | 5.30 a | 0.00 c | 600.00 ab | 27.83 bcd | 35.13 a | 10.15 a |
| | JD27 | 225N | 18.87 c | 5.20 a | 0.00 c | 598.13 ab | 26.83 cd | 32.38 b | 10.50 a |
| | JD27 | 300N | 19.53 c | 4.47 a | 0.00 c | 622.00 a | 26.70 d | 35.61 a | 10.32 a |
| | SD19 | 000N | 21.03 b | 4.70 a | 1.27 a | 549.87 b | 32.73 a | 29.13 c | 5.60 e |
| | SD19 | 075N | 22.90 a | 4.93 a | 1.03 a | 571.87 ab | 30.20 ab | 32.37 b | 7.43 cd |
| | SD19 | 150N | 22.80 a | 5.13 a | 0.70 ab | 616.80 a | 28.53 bcd | 34.13 ab | 9.16 ab |
| | SD19 | 225N | 22.70 a | 5.20 a | 0.33 bc | 619.07 a | 27.57 bcd | 35.28 a | 9.43 ab |
| | SD19 | 300N | 23.10 a | 5.23 a | 0.67 abc | 622.27 a | 28.47 bcd | 34.81 ab | 9.22 ab |
| | F value | Varieties (V) | 153.49 ** | 0.01 | 25.57 ** | 0.46 | 13.57 ** | 0.02 | 10.85 ** |
| | F value | Nitrogen (N) | 3.70 * | 0.84 | 4.36 * | 5.28 ** | 3.63 * | 16.19 ** | 31.82 ** |
| | F value | V:N | 0.92 | 1.15 | 0.46 | 0.17 | 3.92 * | 2.15 | 0.12 |
| | JD27 | | 18.89 b | 5.02 a | 0.15 b | 587.76 a | 27.45 b | 33.22 a | 9.07 a |
| | SD19 | | 22.51 a | 5.04 a | 0.80 a | 595.97 a | 29.50 a | 33.14 a | 8.17 b |
| | | 000N | 19.67 b | 4.83 a | 0.93 a | 545.47 c | 29.55 ab | 29.39 c | 5.97 c |
| | | 075N | 20.67 a | 5.05 a | 0.60 ab | 574.73 bc | 29.87 a | 32.85 b | 7.74 b |
| | | 150N | 21.05 a | 5.22 a | 0.35 b | 608.40 ab | 28.18 abc | 34.63 a | 9.66 a |
| | | 225N | 20.78 a | 5.20 a | 0.17 b | 608.60 ab | 27.20 c | 33.83 ab | 9.97 a |
| | | 300N | 21.32 a | 4.85 a | 0.33 b | 622.13 a | 27.58 bc | 35.21 a | 9.77 a |
| 2016 | JD27 | 000N | 17.08 f | 4.73 cd | 2.07 a | 453.77 c | 26.22 b | 31.90 cd | 5.27 d |
| | JD27 | 075N | 19.47 cde | 5.13 ab | 0.79 ab | 540.38 ab | 24.83 bc | 30.50 d | 8.01 bc |
| | JD27 | 150N | 19.72 bcd | 5.23 a | 0.33 b | 559.48 a | 23.68 cd | 36.88 ab | 9.79 a |
| | JD27 | 225N | 18.17 ef | 5.20 a | 1.22 ab | 536.52 ab | 23.37 cd | 36.09 ab | 9.01 ab |
| | JD27 | 300N | 19.13 de | 4.93 abc | 2.03 a | 502.74 bc | 23.20 d | 35.08 bc | 8.99 ab |
| | SD19 | 000N | 20.43 abc | 4.58 d | 2.40 a | 477.51 c | 27.93 a | 29.40 d | 4.57 d |
| | SD19 | 075N | 21.19 a | 4.65 cd | 2.15 a | 538.14 ab | 25.62 b | 30.87 d | 7.02 c |
| | SD19 | 150N | 20.95 ab | 4.70 cd | 1.32 ab | 547.97 ab | 23.68 cd | 38.62 a | 7.48 c |
| | SD19 | 225N | 21.50 a | 4.80 bcd | 1.57 ab | 579.77 a | 23.88 cd | 38.04 ab | 8.06 bc |
| | SD19 | 300N | 21.40 a | 4.92 abc | 1.67 ab | 557.41 ab | 23.57 cd | 37.26 ab | 7.54 c |
| | F value | Varieties (V) | 74.81 ** | 24.07 ** | 2.53 | 3.89 | 4.71 * | 1.28 | 28.25 ** |
| | F value | Nitrogen (N) | 4.79 ** | 3.50 * | 1.99 | 9.32 ** | 20.15 ** | 22.84 ** | 33.08 ** |
| | F value | V:N | 2.38 | 2.39 | 0.8 | 1.35 | 0.86 | 1.76 | 1.39 |
| | JD27 | | 18.71 b | 5.05 a | 1.29 a | 518.58 a | 24.26 b | 34.09 a | 8.22 a |
| | SD19 | | 21.09 a | 4.73 b | 1.82 a | 540.16 a | 24.94 a | 34.84 a | 6.94 b |
| | | 000N | 18.76 b | 4.66 b | 2.23 a | 465.64 b | 27.07 a | 30.65 b | 4.92 c |
| | | 075N | 20.33 a | 4.89 a | 1.47 ab | 539.26 a | 25.23 b | 30.68 b | 7.52 b |
| | | 150N | 20.33 a | 4.97 a | 0.82 b | 553.73 a | 23.68 c | 37.75 a | 8.64 a |
| | | 225N | 19.83 a | 5.00 a | 1.39 ab | 558.15 a | 23.62 c | 37.06 a | 8.54 a |
| | | 300N | 20.27 a | 4.92 a | 1.85 ab | 530.07 a | 23.38 c | 36.17 a | 8.27 ab |

Different lowercase letters are significant difference at $p < 5\%$ on treatment, ** $p < 0.01$, * $p < 0.05$.

Regression analysis was conducted on fertilizer application and maize grain yield, and the univariate quadratic regression equation was obtained. Table 17 shows that in 2015, the best nitrogen level of JD27 is 248.44 kg ha⁻¹, and the best grain yield is 10.55 t ha⁻¹; the best nitrogen level of SD19 is 237.47 kg ha⁻¹, and the best grain yield is 9.50 t ha⁻¹. In 2016, the best nitrogen level of JD27 was 204.88 kg ha⁻¹, and the best grain yield was 9.68 t ha⁻¹. Also, the best nitrogen level of SD19 was 212.56 kg ha⁻¹, and the best grain

yield was 8.05 t ha$^{-1}$. The comparative analysis showed that maize yield was higher under medium and high nitrogen levels but the yield was inhibited by excessive nitrogen.

**Table 17.** Regression analysis of maize yield under different nitrogen levels.

| Year | Varieties | Binomial Fitting Equation | $R^2$ | Optimum Nitrogen (kg N ha$^{-1}$) | Optimum Yield (t ha$^{-1}$) |
|---|---|---|---|---|---|
| 2015 | JD27 | $y = 6.1994 + 0.0350x - 7.0440*10^{-5}x^2$ | 0.78 | 248.44 | 10.55 |
| | SD19 | $y = 5.5304 + 0.0334x - 7.0324*10^{-5}x^2$ | 0.90 | 237.47 | 9.50 |
| 2016 | JD27 | $y = 5.3759 + 0.0420x - 1.0250*10^{-4}x^2$ | 0.78 | 204.88 | 9.68 |
| | SD19 | $y = 4.7057 + 0.0315x - 7.4950*10^{-4}x^2$ | 0.86 | 212.56 | 8.05 |

## 4. Discussion

Root is a vital organ to fix plants and absorb and transport water and nutrients from soil [43]. Root length, projection area, pixel area, total surface area, total volume and average diameter are important morphological indicators of root development. In particular, root length is the most commonly used index to evaluate root absorption function. Root surface area can reflect the degree of direct contact with soil, and its size directly affects root absorption of soil nutrients. The total root volume is an important part of root architecture, which directly determines the power of root absorption [44]. The root physiological characteristics and morphology was closely related to plant nitrogen uptake [45,46]. Plant dry matter accumulation, plant nitrogen accumulation, leaf area and root amount were significantly positively correlated [47–49]. And the increasing the life span and penetration ability of root after anthesis could enhance the nitrogen absorption capacity of plants and improve the nitrogen absorption efficiency [50]. Nitrogen use efficiency depends on the range and activity of nitrogen uptake by roots [51–53]. The results showed that the root length, projected area, total surface area and total volume increased first, and decreased with the growth period, and reached the highest at silking stage. The average diameter of root kept stable or decreased slowly with the growth period. When the amount of nitrogen is suitable, the morphological and physiological characters of root system can reach the best state, which is helpful to improve the yield of maize and plant lodging resistance. In different growth stages of maize, the amount of nitrogen levels affected root growth in varying degrees. This study found that the root length, projection area, pixel area, total surface area, total volume and average diameter of JD27 and SD19 varied with the increase of nitrogen levels at different stages (V13, R1, R3 and R5). Although higher nitrogen levels could maintain the later root amount, it would increase lodging probability in the later stage of the variety. In addition, with the increase of nitrogen levels, the increasing trend of root indexes of SD19 was more obvious, especially under high nitrogen, while JD27 was more tolerant to low nitrogen.

Maize yield is determined by photosynthetic production and the transportation and distribution of Assimilates in the plant. The basic way to obtain high yield is to improve the dry matter production capacity and the transfer capacity of dry matter to grain after anthesis [54–56]. Dry matter accumulation during growth period is the basis of crop yield formation. In a certain range, grain yield is determined by the level of dry matter accumulation [57,58]. Previous studies have suggested that fertilization and other management measures can significantly improve the accumulation of dry matter and promote the movement, redistribution and grain formation of matter, but the dry matter weight will decrease when the appropriate amount of fertilizer is exceeded [59,60]. Previous studies found that dry matter accumulation of maize showed a "S" curve change, which can be fitted by commonly used logistic equation [17]. The results showed that with the progress of maize growth and development, the dry matter accumulation rate of the two maize varieties at different growth stages showed a trend of first increasing and then decreasing. The establishment of vegetative organs such as stems and leaves before anthesis provides the basis for yield, and the accumulation of photosynthetic metabolites from flowering to maturity is the key to yield formation [61]. Previous studies have suggested that the drier matter is

distributed to the stem in vegetative growth stage, the less leaf area is expanded, and the distribution to storage organs is reduced in reproductive growth stage [62]. In this study, the dry matter distribution ratio of stem and leaf sheath increased first and then decreased with the growth and development process of maize, and the dry matter distribution ratio of leaf decreased gradually with the growth and development process of maize, which indicated that low nitrogen level was not conducive to the accumulation of dry matter and could not meet the needs of vegetative organ growth and development, which made the dry matter transfer to vegetative organs in the late growth and development stage [63]. In this study, the proportion of grain dry matter allocation increased gradually with the growth and development of maize, and the dry matter accumulation under different nitrogen levels mainly came from the photosynthesis after anthesis. Increasing the amount of nitrogen fertilizer can promote the transport of dry matter to grain and improve the efficiency of dry matter transport to grain after anthesis, but excessive application will lead to the decline of transport efficiency [64]. Therefore, higher post anthesis dry matter accumulation capacity and its contribution rate to grain are the physiological basis for improving crop yield [65]. This study also showed that nitrogen played an important role in the increase of dry matter assimilation after anthesis. The appropriate amount of nitrogen could promote the transport and redistribution of dry matter before anthesis and excessive nitrogen levels would inhibit this effect.

Nitrogen is the leading factor affecting crop yield [66], and it is an important nutrient element for crop growth and yield formation. Previous studies have shown that nitrogen application is an effective way to increase yield and income. The grain number per spike of Zhengdan 958 and Nongda 108 is basically the same with the increase of nitrogen levels [67]; the level of nitrogen application affects the 1000 grain weight firstly [68]; insufficient or excessive nitrogen application increases the number of abortive grains, thus affecting the yield [69–71]. Our results showed that: (1) With the increase of nitrogen levels, the number of grains per ear, ear length and ear diameter of maize were significantly improved, and the number of grains per ear of SD19 and JD27 were basically the same, but there was no significant difference between varieties. The number of grains per ear in nitrogen rich treatment was significantly higher than that in nitrogen deficient treatment; (2) With the increase of nitrogen levels, the 100-grain weight of maize showed an increasing trend, and the 100-grain weight in medium and high nitrogen rich treatment was significantly higher than that in nitrogen deficient treatment. The results showed that increasing nitrogen levels was beneficial to increase dry matter accumulation and plumpness of grain; (3) With the increase of nitrogen levels, bald tip length decreased first and then increased. Both nitrogen deficiency and nitrogen enrichment could promote bald tip length. Appropriate nitrogen levels could make nitrogen absorbed by plant distribute to grain reasonably and efficiently, and improve nitrogen use efficiency; (4) With nitrogen levels increasing, there was no significant difference in yield under medium and high nitrogen enrichment. The yield of maize was higher under medium and high nitrogen levels, but the yield was lower under excessive nitrogen levels. Further regression analysis showed that high yield could be obtained easily under medium and high nitrogen levels.

## 5. Conclusions

Nitrogen plays an important role in the increase of dry matter assimilation after anthesis. An appropriate amount of nitrogen can promote the transport and redistribution of dry matter before anthesis and excessive nitrogen levels can inhibit this process effect. Increasing nitrogen fertilizer was helpful to promote the formation of yield. Under the treatment of medium and high nitrogen fertilizer, maize was easy to obtain higher yield, but excessive nitrogen fertilizer inhibited the increase of yield. This study provides theoretical guidance for the breeding of high yield and high efficiency maize varieties and the regulation of high yield and high efficiency cultivation techniques.

**Author Contributions:** Y.Y. performed the entire experiment, analysed the results, and drafted the manuscript. C.L. and W.G. designed the entire experiment and were in charge of manuscript revisions. W.G. and C.Q. interpreted the results and prepared the manuscript. All authors have read and agreed to the published version of the manuscript.

**Funding:** This research was funded by Excellent Youth Foundation of Heilongjiang Province of China (Grant No. JC2017008) and China Agriculture Research System (Grant No. CARS-02-34).

**Institutional Review Board Statement:** The study in the paper did not involve humans or animals.

**Informed Consent Statement:** The study in the paper did not involve humans or animals.

**Data Availability Statement:** The data presented in this study are available on request from the corresponding author.

**Conflicts of Interest:** The authors declare no conflict of interest.

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
