# Peer review of "Responses of Root Characteristic Parameters and Plant Dry Matter Accumulation, Distribution and Transportation to Nitrogen Levels for Spring Maize in Northeast China"

_agriculture, doi:10.3390/agriculture11040308_

Round 1
Reviewer 1 Report
The authors presented a lot of really interesting data. However, authors must provide answers to questions that arose during the revision. Therefore:
It is not clear in which rows the plants were taken for analysis (V13, R1, R3, and R5). The authors stated that at the mature stage, the middle two rows of each plot harvested.
Did the authors use plants from the edge rows (edge effect)?
How did the authors determine that the plants are in a certain stage (e.g. silking stage)?
Are the data for ear weight, and grain weight, were adjusted to a moisture content of 14%?
It is not clear why authors sometimes use the term extremely significant for (P<0,05) and sometimes for (P<0,01)?
In the following rows is necessary to fix the following:
434 Please?
551 000N
527 and 528 per panicle or per ear?
590 Table 16 or Table 17?
639 The results showed or the results are given from this study showed?
The authors did not explain how weather conditions affected the results obtained.
The authors in the discussion did not compare the results which they obtained with data obtained by other authors.
Conclusions need to be refined.
Reviewer 2 Report
My comments on the manuscript “Responses of Root Characteristic Parameters and Plant Dry Matter Accumulation, Distribution and Transportation to Nitrogen Levels for Spring Maize in Northeast China”, which has been submitted to Agriculture journal, are presented below.
The manuscript is very interesting. Below I only provide suggestions for the Authors of the work to consider.
- What technology of soil cultivation was used?
- Please complete the data on agricultural technology – previous crop and level of chemical protection of the crop [seed dressing, herbicides, fungicides, insecticides].
- Describing the chemical composition of the soil, the Authors cite Table 1 regarding meteorological conditions. However, there is no short description for the table with meteo conditions and its correct citation.
- I consider part of the descriptions for Figures 1, 5, 6 in a language other than English to be unnecessary. In my opinion, it would be better to enter the description only in English.
- It seems as if an unfinished pattern is included within Figure 2. At the same time, the source given for this formula "34" is not available under this number doi. Please check.
- Please standardize the units of measurement used in the description. The Authors give both t ha-1 [Fig.4, Fig5 - In the description, the Authors used kg hm-2 lines 353; 354]. and t hm-2 [Fig.6].
In my opinion, the remaining elements of the manuscript do not raise any objections.
Round 2
Reviewer 1 Report
Dear authors thank you for your cooperation. Also, thank you for adding, changing, and explaining what was needed. I have no more requests and in my opinion, your manuscript is ready for publication.
Author Response
Thank you very much for your suggestions. We have revised and improved the manuscript comprehensively.